# Drivers of macroinvertebrate community structure in unmodified streams

Jonathan D. Tonkin

Department of Environmental Science, Xi'an Jiaotong-Liverpool University, Dushu Lake
Higher Education Town, SIP, Suzhou, Jiangsu Province, China
Department of River Ecology and Conservation, Senckenberg Research Institute and Natural
History Museum, Clamecystrasse, Gelnhausen, Germany
Biodiversity and Climate Research Centre, Frankfurt am Main, Germany

## ABSTRACT

Often simple metrics are used to summarise complex patterns in stream benthic ecology, thus it is important to understand how well these metrics can explain the finer-scale underlying environmental variation often hidden by coarser-scale influences. I sampled 47 relatively pristine streams in the central North Island of New Zealand in 2007 and (1) evaluated the local-scale drivers of macroinvertebrate community structure as well as both diversity and biomonitoring metrics in this unmodified landscape, and (2) assessed whether these drivers were similar for commonly used univariate metrics and multivariate structure. The drivers of community metrics and multivariate structure were largely similar, with % canopy cover and resource supply metrics the most commonly identified environmental drivers in these pristine streams. For an area with little to no anthropogenic influence, substantial variation was explained in the macroinvertebrate community (up to 70% on the first two components of a partial least squares regression), with both uni- and multivariate approaches. This research highlights two important points: (1) the importance of considering natural underlying environmental variation when assessing the response to coarse environmental gradients, and (2) the importance of considering canopy cover presence when assessing the impact of stressors on stream macroinvertebrate communities.

## INTRODUCTION

Streams are an important biodiversity reservoir within landscapes and benthic macroinvertebrates account for a large amount of this biodiversity. Many factors can influence stream macroinvertebrate diversity ranging from patch- to landscape-scale drivers (*Vinson & Hawkins, 1998*; *Heino, 2009*), including temperature and altitude (*Jacobsen, Schultz & Encalada, 1997*), disturbance (*Death & Winterbourn, 1995*; *Tonkin & Death, 2012*), and past (*Harding et al., 1998*) and current land use (*Allan, 2004*). However, there is little consensus regarding any specific overriding influences on structure and diversity, but that they are a complex function of multiple environmental factors at multiple spatial and temporal scales (*Poff, 1997*; *Heino, 2009*). Catchment features generally exert influence on local-scale environmental features in stream systems (*Johnson & Gage, 1997*;

Corresponding author
Jonathan D. Tonkin,
jonathan.tonkin@senckenberg.de

*Johnson et al., 1997*), just as regional species pools filter local macroinvertebrate colonisers (*Poff, 1997*; *Heino, Muotka & Paavola, 2003*; *Tonkin et al., 2014*). Yet, the importance of these larger-scale influences may be dependent on some form of human pressure such as land-use change being present (*Heino, Mykrä & Kotanen, 2008*), and often their influences are weak (*Hawkins et al., 2000*).

Both in biodiversity assessments and biomonitoring programs, these complex and diverse communities are often reduced to a single number to represent the entire community, such as a multitude of metrics representing species diversity (*Magurran, 2004*), biotic integrity (e.g., *Kerans & Karr, 1994*), and the effects of organic enrichment (*Stark, 1985*; *Smith et al., 1999*). However, metrics respond in different ways to environmental stressors (*Yuan & Norton, 2003*). Thus the single-metric approach can overlook important information about the community and its response to stressors (*Reynoldson et al., 1997*; *Collier, 2009*), highlighting the importance of using a variety of indices to indicate or represent the full range of environmental conditions. However, specific metrics may lack sensitivity to underlying variation in community structure resulting from other environmental conditions for which they were not designed. In essence, natural underlying environmental variation can obscure patterns along these stressor gradients. Other approaches, including multimetric, multivariate and predictive approaches (e.g., *Clarke, Wright & Furse, 2003*; *Collier, 2009*; *Kerans & Karr, 1994*), may thus be more appropriate.

One approach to assess the extent of this issue is to explore how well these metrics respond to environmental variation within pristine environments, where human stressors are largely absent. Previous studies have highlighted the importance of local-scale factors in shaping stream macroinvertebrate communities (*Death & Joy, 2004*; *Astorga et al., 2011*). Accordingly, I aimed to (1) evaluate the local-scale drivers of macroinvertebrate community structure and diversity patterns in streams across a relatively unmodified landscape, and (2) assess whether these drivers were similar for commonly used univariate metrics and multivariate structure. To do this, I calculated several commonly applied univariate metrics, ranging from simple diversity indices to those developed for responding to organic enrichment (e.g., Macroinvertebrate Community Index; MCI; *Stark, 1985*), and assessed the environmental factors explaining variation in these metrics and multivariate structure. Theoretically, community metrics should respond similarly to more complex multivariate analyses, but with variation in response depending on the metric's designed purpose. Therefore, I predicted that despite more variation being explained in the multivariate analyses, the main environmental influences would be similar for both univariate and multivariate approaches. Given the pristine nature of the streams sampled, I hypothesised the limited environmental gradients present would lead to low levels of variation explained.

## MATERIAL AND METHODS

### Study sites
The Tongariro National Park in the central North Island of New Zealand comprises a central volcanic massif of three mountains of predominantly andesitic geology and the

**Peer**J ___________________________________________

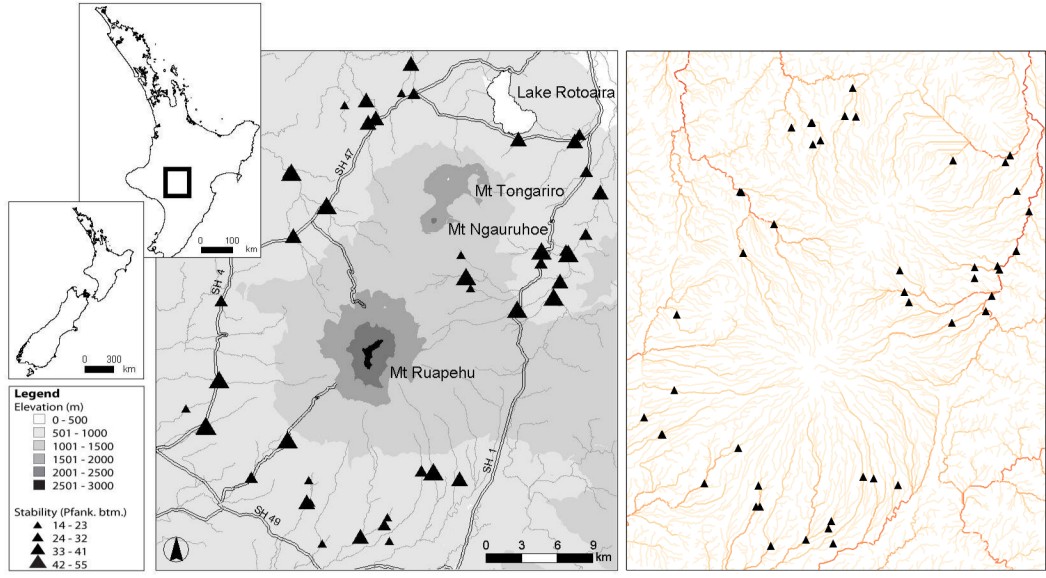

**Figure 1 Map of 47 study sites.** Location of the 47 study sites in the Tongariro National Park, New Zealand. Plate on right is colour coded from light to dark based on increasing stream order to indicate the number of major drainages that were sampled. Pfank. btm, bottom component of the Pfankuch stability index.

Tihia-kakaramea volcanic massif to the north. For sampling, 47 first- to sixth-order streams and rivers were selected from within the Tongariro National Park that were subjected to minimal human interference by excluding sites with greater than 10% catchment pastoral land use. All sites had a minimum of 90% volcanic hard sedimentary geology. Ten of the 47 sites had flows modified by hydroelectric dams, but as these were run-of-river type dams, they were not able to hold back floods and thus kept relatively natural flow regimes. The flow regimes of the remainder of sites were unmodified and ranged from runoff-fed streams to stable spring-fed streams. Eight sites were situated within *Pinus radiata* plantation forestry, but these sites were limited to mature forest, which has been found to have similar macroinvertebrate communities to native forest (*Quinn et al., 1997*; *Quinn, Boothroyd & Smith, 2004*). These sites belong to eight fifth order drainages (Fig. 1). For a more detailed description of the study sites, see *Tonkin, Death & Collier (2013)*.

## Biological collections
### *Macroinvertebrates*
All sampling was performed on one occasion, between early February and late April 2007. Macroinvertebrates were sampled by taking five 0.1-m$^2$ Surber samples (250 μm mesh, but later sieved to 500 μm) from random locations from riffles throughout ca. 50-m study reaches. Samples were preserved in 10% formalin before identification to the lowest possible taxonomic level in the laboratory, using available keys (e.g., *Towns & Peters, 1996*; *Winterbourn, Gregson & Dolphin, 2000*). Where certain taxa could not be identified to species level (e.g., Chironomidae and Oligochaeta), I identified them to morphospecies.

Several indices were calculated to summarise different aspects of the macroinvertebrate community, ranging from diversity measures to biomonitoring metrics developed as indicators of organic pollution. These are as follows: number of individuals (density), number of taxa (richness), Macroinvertebrate Community Index (MCI) (*Stark, 1985*) and its quantitative variant the QMCI (*Stark, 1993*), percent of Ephemeroptera, Plecoptera and Trichoptera taxa (%EPT taxa), percent of EPT individuals (%EPT individuals), the number of EPT taxa (EPT richness), and Margalef's diversity index (*Clifford & Stephenson, 1975*). The MCI and QMCI were developed to indicate organic enrichment in stony bottom streams and are similar to the Hilsenhoff Biotic Index (*Hilsenhoff, 1987*).

I did not include fish in this study for two reasons: (1) In the flood prone streams of Tongariro National Park, it is unlikely that predation biomass will ever be great enough to have a strong influence. The only fish likely to be present in these streams are juvenile rainbow trout (*Oncorhynchus mykiss*), but densities would not reach high enough levels to exert clear pressure on macroinvertebrate communities. (2) It is likely that the predation effect, while minor, will be relatively even across all sites.

### *Periphyton*

Periphyton biomass was assessed from measures of chlorophyll *a* taken from five stones (mean area: 60 cm$^2$) at each site. These were collected randomly from riffles within the sampling reach and transported in the dark and on ice prior to being frozen. Chlorophyll *a* was extracted by using 90% acetone at 5 °C for 24 h in the dark. Absorbances were then read at 750, 665 and 664 nm on a Varian Cary 50 conc UV-Visible Spectrophotometer (Varian Australia Pty Ltd, Mulgrave, Australia) and converted to pigment concentration as per *Steinman & Lamberti (1996)*. These values were then estimated and corrected for stone surface area using *Graham, McCaughan & McKee (1988)* and halved to account for the fact that only half the stone surface is available for periphyton growth.

Recent studies have demonstrated the applicability of rapid assessment methods of periphyton cover (*Kilroy et al., 2013*; *Tonkin, Death & Barquin, 2014*). Therefore, the percentage of periphyton cover was visually assessed and broken into four categories. These were: bare (no cover), thin films (0–1 mm), mats (>1 mm) and filamentous algae. Percent bryophyte and macrophyte cover was also assessed along these transects.

## Physicochemical sampling

Physicochemical assessment was performed concurrently with biological sampling and sampled variables can be found in Table 1. Bed stability was assessed using the substrate component of the Pfankuch stability index (*Pfankuch, 1975*), which consists of rock angularity, brightness, packing, percent stable materials, scouring and amount of clinging vegetation.

Substrate size composition was assessed using the 'Wolman Walk' method, by selecting and measuring (beta axis) 100 stones at 1-m intervals 45° to the stream bank in a zigzag manner (*Wolman, 1954*) and grouping into Wentworth scale size classes (*Wentworth, 1922*; *Cummins, 1962*). The percentage of these classes was then converted into a single substrate size index (SI) by summing their midpoint values weighted by their proportion (I assigned

**Table 1 Summary statistics of macroinvertebrate metrics and environmental variables.** Summary statistics of the eight macroinvertebrate metrics and 24 environmental variables sampled from 47 streams in the Tongariro National Park, New Zealand, 2007.

| Variable | Min | Max | Mean | S.E. |
|---|---|---|---|---|
| Richness | 4.00 | 45.00 | 27.28 | 1.17 |
| Number of individuals | 15.00 | 5978.00 | 1486.30 | 168.06 |
| MCI | 90.00 | 134.50 | 113.50 | 1.48 |
| QMCI | 3.07 | 7.75 | 5.95 | 0.18 |
| %EPT taxa | 35.29 | 73.08 | 54.75 | 1.17 |
| %EPT individuals | 7.85 | 90.85 | 58.83 | 3.52 |
| EPT richness | 2.00 | 27.00 | 15.19 | 0.80 |
| Margalef's | 0.99 | 6.08 | 3.72 | 0.15 |
| Chlorophyll $a$ ($\mu$g cm$^{-2}$) | 0.03 | 5.02 | 1.87 | 0.19 |
| Conductivity ($\mu$S cm$^{-1}$) | 40.00 | 298.00 | 112.80 | 8.54 |
| Depth (cm) | 5.70 | 52.20 | 27.19 | 1.61 |
| Embededness | 1.00 | 3.00 | 2.06 | 0.10 |
| Pfankuch | 14.00 | 55.00 | 33.55 | 1.66 |
| pH | 6.70 | 8.70 | 7.95 | 0.07 |
| Slope (m 10 m$^{-1}$) | 1.04 | 12.80 | 4.30 | 0.31 |
| Substrate heterogeneity | 1.44 | 2.30 | 1.92 | 0.03 |
| Substrate size index | 43.67 | 254.64 | 143.97 | 7.20 |
| Temperature (°C) | 6.60 | 17.60 | 10.79 | 0.31 |
| Velocity (m s$^{-1}$) | 0.16 | 1.46 | 0.78 | 0.05 |
| Width (m) | 1.43 | 30.00 | 9.35 | 1.23 |
| % Bryophyte cover | 0.00 | 90.00 | 13.57 | 3.53 |
| % CPOM cover | 0.00 | 60.00 | 7.36 | 1.67 |
| % Debris jam | 0.00 | 60.00 | 3.13 | 1.36 |
| % Filamentous cover | 0.00 | 28.00 | 2.51 | 0.69 |
| % Film cover | 0.00 | 90.00 | 46.98 | 4.22 |
| % Macrophyte cover | 0.00 | 20.00 | 0.43 | 0.43 |
| % Mat cover | 0.00 | 85.00 | 14.26 | 3.51 |
| % Native forest | 0.00 | 100.00 | 39.15 | 5.72 |
| % Native scrub | 0.00 | 100.00 | 54.26 | 5.35 |
| % Overhead cover | 0.00 | 90.00 | 31.64 | 4.65 |
| % Planted forest | 0.00 | 80.00 | 6.38 | 2.61 |
| % Undercut | 0.00 | 40.00 | 3.28 | 1.01 |

**Notes.**

MCI, Macroinvertebrate Community Index; QMCI, Quantitative MCI; %EPT, %Ephemeroptera, Plecoptera and Trichoptera.

bedrock as 400 mm for use in the calculation) (*Quinn & Hickey, 1990*). Stream-slope was assessed as the height drop over 10–100-m sections depending on the size of the stream. Substrate heterogeneity was calculated using the Shannon diversity index, and embeddedness graded on a three-point scale from loose to tight.

Conductivity, temperature and pH were spot-measured using a Eutech ECScan and pHtestr2 (Eutech Instruments, Singapore) respectively. Depth and current velocity were measured using a Marsh McBirney flomate current meter (Marsh McBirney, Frederick,

Maryland) at five equidistant points along the thalweg, and width at three points, of each study reach.

The percentage of coarse particulate organic matter (CPOM) was visually assessed in the substrate along five transects throughout each reach. The percentage of debris jams and undercut banks were visually assessed along the entire study reach, as well as the percentage of overhead cover shading the stream. The percentage of riparian vegetation was visually assessed along each study reach broken into the following categories: native forest, native scrub, planted forest, pasture and bare ground.

Finally, to assess the link between overhead cover and catchment bare ground and tussock cover, catchment land-use for each site was extracted from the River Environment Classification (REC) (*Snelder & Biggs, 2002*), and % bare ground and % tussock cover was combined for analyses.

## Data analysis

All analyses were performed using R version 2.15.2 (*R Core Team, 2013*). To assess the strength of the link between stream width and % overhead cover, and whether this link was confounded by elevation, Pearson's correlation was performed using the cor.test() function. Further, to test if this link was reflected by the catchment cover of tussock and bare ground, overhead cover was correlated with the combined tussock and bare ground percentage.

Spatial autocorrelation was examined using a Mantel test based on Pearson's product-moment correlation and 999 permutations, with the mantel() function in the package Vegan (*Oksanen et al., 2011*). This was performed by comparing geographic with environmental and macroinvertebrate dissimilarity matrices calculated using the function vegdist() in Vegan. Geographic and environmental matrices were calculated using Euclidian distances and macroinvertebrate matrices were calculated using Bray-Curtis distances on $\log (x + 1)$ transformed macroinvertebrate data. Geographic data was simply the NZTM (New Zealand Transverse Mercator) easting and northing coordinates.

To explore the best set of environmental drivers of community metrics, partial least squares regression (PLSR) was performed using the plsr() function in the pls package (*Mevik & Wehrens, 2007*). First, the environmental variables were standardised using the scale() function. PLSR was then performed using the full set of 24 standardised environmental variables and models were validated using leave-one-out cross validation. This was carried out for each of the eight macroinvertebrate metrics separately to get the best possible prediction for each individual metric and the number of components was limited to two. For the cross-validated models, the root mean square error of the estimate was calculated to evaluate the model.

For all multivariate analyses, the raw invertebrate data was $\log (x + 1)$ transformed to reduce heteroscedasticity. To select the best subset of environmental variables explaining variation in the multivariate macroinvertebrate community structure, BIO-ENV (*Clarke & Ainsworth, 1993*) was performed with bioenv() in the Vegan package. This function selects the best subset of environmental variables by maximising the correlation between

environmental (using Euclidean distances) and community distance matrices. Spearman rank correlations were used to correlate the matrices and Bray-Curtis distances for the community dataset.

To visually assess the multivariate structure of the macroinvertebrate community, non-metric multidimensional scaling (nMDS) ordination was performed using the metaMDS() function in the Vegan package. Again, Bray-Curtis distances were used and the number of axes was limited to two. To examine the gradient of effect of the variables identified in BIO-ENV, smooth surface thinplate splines were fitted using the ordisurf() function in Vegan. This procedure uses generalized additive models (GAMs) to overlay a smoothed response surface, which allows a more detailed interpretation than a simple linear vector.

## RESULTS

### Environmental variables

Conductivity ranged from 40 to 298 µS cm$^{-1}$, averaging 112.8 µS cm$^{-1}$ (Table 1). Spot temperature ranged from 6.6 to 17.6 °C, with a mean of 10.79 °C. Velocity ranged from 0.16 to 1.46 m s$^{-1}$, with a mean of 0.78 m s$^{-1}$ (Table 1). Mean depth was 27.19 cm and varied between 5.7 and 52.2 cm. Substrate size varied between 43.67 and 254.64 mm, with a mean of 143.97 mm.

Mean chlorophyll *a* was 1.87 µg cm$^{-2}$ and ranged from 0.03 to 5.02 µg cm$^{-2}$ (Table 1). Thin films were the dominant growth form of periphyton with a mean of 46.98% cover, followed by mats with 14.26% cover (Table 1). Filamentous algae was rarely present with a mean of 2.51% cover. Bryophytes were highly variable ranging from 0 to 90% cover, with a mean of 13.57% cover. Environmental data were not spatially autocorrelated ($r = 0.03$, $p = 0.2$), but there was a spatial association for the macroinvertebrate data ($r = 0.08$, $p = 0.02$).

### Univariate metrics

A total of 97 taxa was collected from the 47 sites in this study. Insects dominated all the sites and were the most taxonomically rich, with 35 caddisfly (Trichoptera) taxa, 22 Diptera, 14 mayflies (Ephemeroptera), and eight stoneflies (Plecoptera).

The pooled number of invertebrates in the benthos ranged from 15 to 5978 individuals 0.5 m$^{-2}$ and the number of taxa collected at each site ranged from 4 to 45 taxa 0.5 m$^{-2}$ (Table 1). MCI ranged from 90 to 134.5, averaging 113.5, and QMCI ranged between 3.07 and 7.75, averaging 5.95. Percent EPT taxa and individuals averaged approximately the same, between 55 and 59%, but the range was much greater for %EPT individuals (7.85–90.85) than %EPT taxa (35.29–73.08; Table 1). There were, on average, 15 EPT taxa per site, ranging between 2 and 27 taxa.

### PLSR

*Overall*

The first two components of the partial least squares regressions were able to explain between 41 and 70% of the variation in the eight macroinvertebrate metrics (Table 2).

**Table 2  Partial least squares regression summary statistics.** Results of partial least squares regression (PLSR) for each of the eight macroinvertebrate metrics collected from 47 streams in the Tongariro National Park, 2007. The first two columns are the results using leave-one-out cross validation and the remaining columns are those using the full set of data.

| Metric | Cross validated | | Training dataset (% variance explained) | | | |
|---|---|---|---|---|---|---|
| | RMSE | | Environmental variables | | Macroinvertebrate metrics | |
| | Comp 1 | Comp 2 | Comp 1 | Comp 2 | Comp 1 | Comp 2 |
| Richness | 6.945 | 6.241 | 14.84 | 23.93 | 49.06 | 69.18 |
| Log (abundance) | 0.4291 | 0.4221 | 9.993 | 21.1 | 49.153 | 64.12 |
| MCI | 9.005 | 9.771 | 16.7 | 24.81 | 42 | 52.01 |
| QMCI | 1.26 | 1.351 | 13.7 | 24.49 | 29.27 | 40.64 |
| %EPT taxa | 7.541 | 7.753 | 14.57 | 24.69 | 40.65 | 52.04 |
| %EPT individuals | 23.36 | 24.09 | 12.47 | 24.65 | 38.21 | 47.96 |
| EPT richness | 4.512 | 4.071 | 14.99 | 24.46 | 53.55 | 70.38 |
| Margalef's index | 0.9097 | 0.837 | 15.41 | 23.9 | 48.13 | 68.29 |

**Notes.**
RMSE, root mean square error of estimate; MCI, Macroinvertebrate Community Index; QMCI, Quantitative MCI; %EPT, %Ephemeroptera, Plecoptera and Trichoptera.

These components explained much less variation in the environmental data, ranging between 21 and 25%. Taxonomic richness ($r^2 = 0.69$), EPT richness ($r^2 = 0.70$) and Margalef's index ($r^2 = 0.68$) were those best predicted by the environmental variables, whereas QMCI was least successfully predicted by the two components ($r^2 = 0.41$; Table 2). Component 1 variation explained ranged from 29% (QMCI) to 54% (EPT richness), with taxonomic richness ($r^2 = 0.49$), density ($r^2 = 0.49$) and Margalef's ($r^2 = 0.48$) closely following. Percent overall variation explained for component 2 ranged from 10% (%EPT individuals) to 20% (both taxonomic richness and Margalef's; Table 2).

### Key variables

The variables with the consistently highest (or most negative) loadings on either component across all eight macroinvertebrate metrics were % overhead cover, % bryophyte cover, % debris jam and depth (Fig. 2).

Percent overhead cover was one of the strongest predictors of the macroinvertebrate metrics (Fig. 2). Overhead cover had a loading greater than 0.4 on all of the first PLS components for all metrics except for density (richness = 0.49, MCI = 0.48, QMCI = 0.48, %EPT taxa = 0.52, %EPT individuals = 0.45, EPT richness = 0.5, Margalef's = 0.5; Fig. 2) and % CPOM cover contributed slightly lower positive loadings to the same component (richness = 0.36, MCI = 0.37, QMCI = 0.37, %EPT taxa = 0.38, %EPT individuals = 0.35, EPT richness = 0.35, Margalef's = 0.36; Fig. 2). On the same components that % CPOM and overhead cover were positively loaded, width exhibited weaker negative loadings between −0.29 and −0.36, as width and overhead cover were negatively correlated ($r = −0.45$, $p = 0.001$). However, overhead cover also declined with elevation ($r = −0.33$, $p = 0.025$), but there was no link between elevation and width ($r = −0.08$, $p = 0.58$). Overhead cover was highly negatively correlated with the percentage of the catchment with bare ground and tussock cover ($r = −0.73$, $p < 0.0001$).

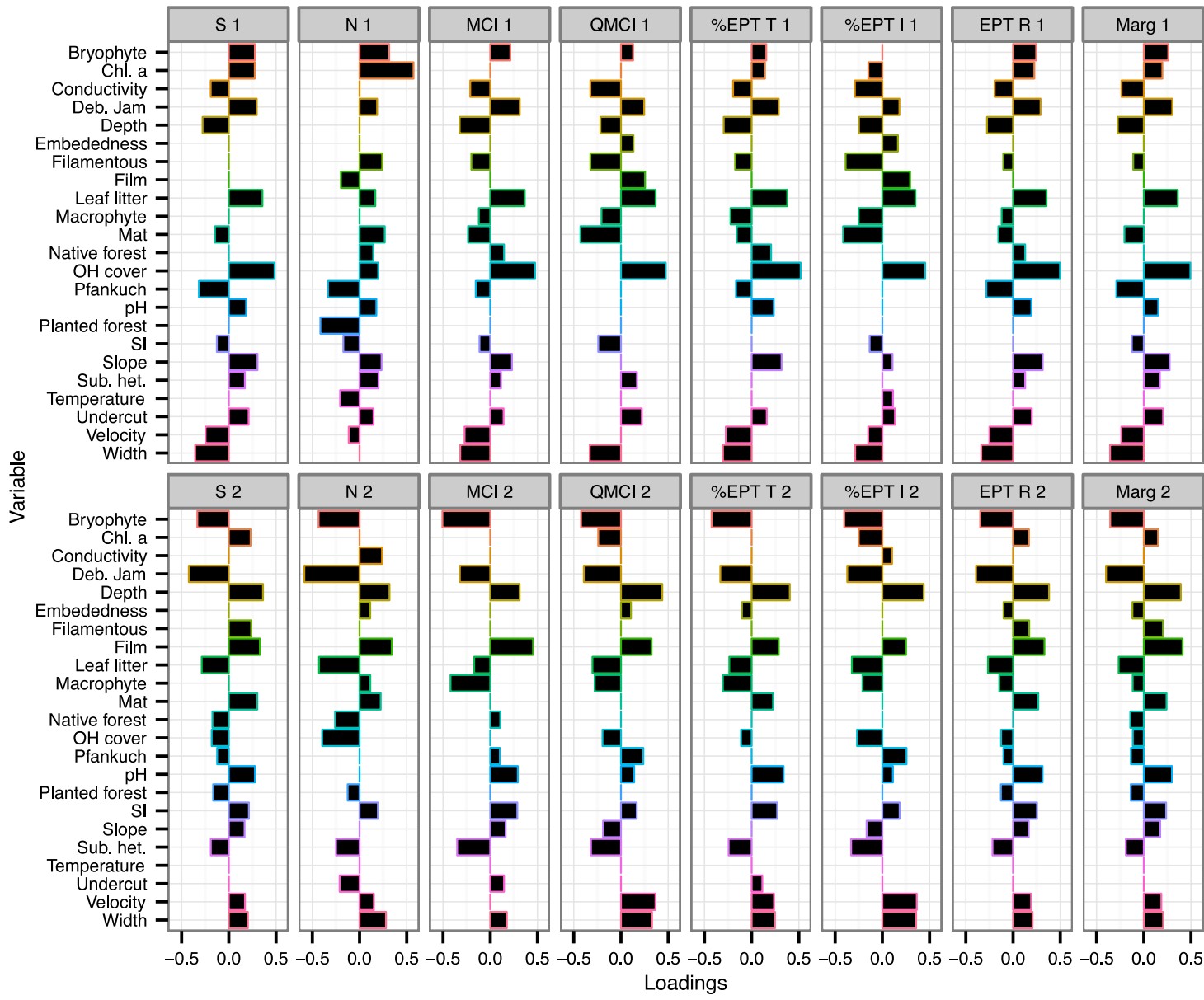

**Figure 2 Loadings of environmental variables on partial least squared regression models.** Loadings plot the contribution of 24 environmental variables to the first two components of eight partial least squares regression (PLSR) models. Individual PLSR models were calculated for the eight macroinvertebrate metrics collected from 47 Tongariro National Park, New Zealand streams between February and April 2007. S, taxonomic richness; N, number of individuals; MCI, Macroinvertebrate Community Index; QMCI, Quantitative MCI; %EPT T, %Ephemeroptera, Plecoptera and Trichoptera taxa; %EPT I, %EPT individuals; EPT R, EPT richness; Marg, Margalef's index. Numbers represent PLSR component number. Chl. a, chlorophyll *a*; Deb. Jam, debris jam; OH Cover, overhead cover; SI, substrate size index; Sub. het., substrate heterogeneity.

Chlorophyll *a* had the strongest loading for either component predicting macroinvertebrate density with a loading of 0.57 on component 1 and on the same component % planted forest contributed −0.42 (Fig. 2). Percent bryophyte cover (−0.43), % debris jam (−0.59) and % CPOM cover (−0.43) all had strong negative loadings on the second component predicting the number of individuals (Fig. 2). As well as a strong

**Table 3 BIO-ENV correlating environmental data and macroinvertebrate community structure.** The best subset of variables selected using BIO-ENV correlating multivariate community structure with (a) all, (b) EPT only and (c) non-EPT taxa collected from 47 streams in the Tongariro National Park, 2007.

| All | | EPT | | Non-EPT | |
|---|---|---|---|---|---|
| Variable | Spearman rho | Variable | Spearman rho | Variable | Spearman rho |
| Bryophyte | 0.375 | Bryophyte | 0.426 | Pfankuch | 0.227 |
| + Planted forest | 0.442 | + Planted forest | 0.483 | + Overhead cover | 0.323 |
| + Pfankuch | 0.480 | + Macrophyte | 0.496 | + Bryophyte | 0.368 |
| + Overhead cover | 0.525 | + Depth | 0.518 | + Debris jam | 0.418 |
| + Debris jam | 0.536 | + Planted forest | 0.546 | + Temperature | 0.430 |
| + Temperature | 0.548 | + Debris jam | 0.555 | + Slope | 0.442 |
| + Chlorophyll *a* | 0.554 | + Temperature | 0.564 | + Conductivity | 0.453 |
| + Depth | 0.562 | + Pfankuch | 0.578 | + Embeddedness | 0.463 |
| | | + Chlorophyll *a* | 0.584 | + Native scrub | 0.468 |
| | | + Overhead cover | 0.586 | + % Films | 0.468 |
| | | | | + Pfankuch | 0.468 |

negative loading on component 2 of the density model, debris jam contributed strong negative loadings on all of the remaining metrics (richness $= -0.42$, MCI $= -0.32$, QMCI $= -0.39$, %EPT taxa $= -0.33$, %EPT individuals $= -0.37$, EPT richness $= -0.39$, Margalef's $= -0.4$; Fig. 2). Moreover, in addition to a positive contribution by % overhead cover on component 1, % periphyton mat cover contributed strongly negatively to predict QMCI ($-0.43$) and %EPT individuals ($-0.42$; Fig. 2).

Percent bryophyte cover also contributed strongly negatively to component 2 of the MCI ($-0.51$), QMCI ($-0.42$), %EPT taxa ($-0.43$) and %EPT individuals ($-0.4$) models (Fig. 2). Percent periphyton film cover only contributed strongly to the second component of MCI (0.45) and Margalef's index (0.41). The Pfankuch stability index contributed negatively to component 1 of taxonomic richness ($-0.32$) and density ($-0.33$), but contributed little to any of the other models (Fig. 2).

## Multivariate

The best model using BIO-ENV selected eight of the 24 environmental variables (% overhead cover, Pfankuch, chlorophyll *a*, depth, temperature, % debris jam, % planted forest, % bryophyte cover) and had a correlation (rho) of 0.562 with the multivariate community structure (Table 3). However, there was little improvement in the link with community structure from the model with four variables (rho $= 0.525$). This model included % overhead cover, Pfankuch, % planted forest and % bryophyte cover.

The nMDS ordination on log $(x + 1)$ transformed macroinvertebrate communities produced a reasonable ordination with a stress of 0.159 (Fig. 3). Overlaying this ordination with individual GAM fitted smooth surfaces of each of the eight variables selected by BIO-ENV using thinplate splines indicates the main influence of these important variables was along nMDS axis 1 (Fig. 3). Sites positively loaded on axis 1, tended to be more

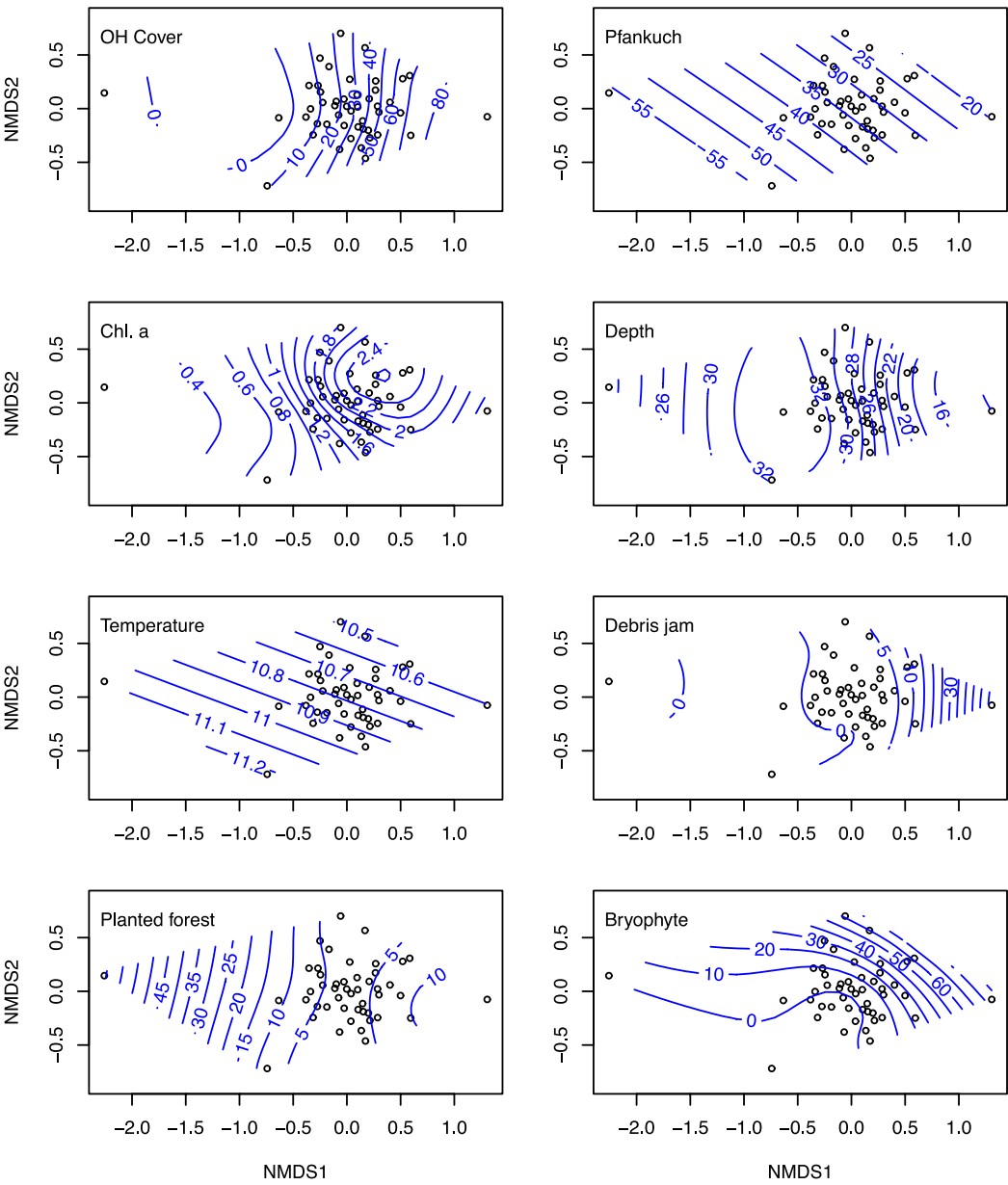

**Figure 3 nMDS ordinations on macroinvertebrate communities of 47 study sites.** Biplots of the non-metric multidimensional scaling ordination on log $(x + 1)$ transformed macroinvertebrate community data collected from 47 streams in the Tongariro National Park, New Zealand, 2007. Individual plots display overlaid smooth surface thinplate splines using generalized additive models (GAMs) for the eight environmental variables selected using BIO-ENV. Numbers on the splines represent value of the specific environmental variable. OH Cover, overhead cover; Chl. a, chlorophyll *a*.

resource rich with greater levels of chlorophyll *a*, bryophytes, overhead cover and more stable (i.e., Pfankuch).

One site (site 32), which had very low abundance (four taxa and 15 individuals collected), loaded much more negatively on axis 1 than any other site (Fig. 3). This site was situated in a plantation forestry area, which may explain the spline loadings of % planted

forest declining along this axis. At the other end of the scale, site 45, which exhibited the highest loading on nMDS 1, was a small spring-brook with a high percentage of bryophyte cover and overhead canopy cover, with a macroinvertebrate community that reflected these physical factors.

Removing this site from the nMDS increased the two-dimensional stress from 0.159 to 0.182. Furthermore, running the BIO-ENV with this site excluded reduced the link between environmental variables and the invertebrate community from rho = 0.562 to rho = 0.533, and the set of explanatory variables remained similar, with the notable exclusion of % planted forest. This BIO-ENV selected nine variables: % overhead cover, Pfankuch, chlorophyll *a*, depth, temperature, debris jam, substrate size, % bryophyte, and % macrophyte. Finally, excluding site 32 from the PLSR on abundance (which was the only invertebrate metric % planted forest influenced strongly) removed the influence of planted forest on density, but also reduced the amount of variation explained by the first two components from 64% to 55%.

Splitting the invertebrate community into EPT and non-EPT taxa indicated similar multivariate structure (Mantel $r = 0.62$, $p = 0.001$), and BIO-ENV revealed similar drivers, with 10 and 11 variables selected for each of EPT and non-EPT data respectively (Table 3).

Of the eight variables explaining multivariate structure, temperature was the only one that contributed little to univariate metric prediction. The biggest loading for temperature on any of the PLS components, was $-0.21$ on component 1 predicting macroinvertebrate density. The remainder of environmental variables contributed to both univariate and multivariate prediction.

## DISCUSSION

### Metric use

Contrary to my hypothesis, local-scale environmental variables were able to explain substantial variation in taxonomic richness of benthic macroinvertebrate communities in these pristine mountain streams with two components of a PLSR (70%). This is a relatively strong explanatory power for a set of streams with a highly constrained range of landscape-scale factors such as geology, climate, land-use and altitude. Nonetheless, there was considerable natural variation in environmental variables measured at these sites, which may explain the strong link with the macroinvertebrate community. In accordance with my primary hypothesis, the environmental drivers of univariate macroinvertebrate metrics and those of multivariate community structure were essentially similar. Where regional differences are present in environmental conditions, the regional species pool is an important driver of local species richness, with the ability to override local scale influences in streams (*Heino, Muotka & Paavola, 2003*; *Tonkin et al., 2014*). However, without these regional differences, the importance shifts back to local-scale drivers, especially if large-scale factors such as geology are limited as they were in the present study.

I used a limited range of community metrics, which could potentially mask important information, given metrics both respond differently to different stressors (*Yuan & Norton, 2003*) and only represent a small part of community dynamics. A multimetric approach

may overcome this reduction of complex relationships by simultaneously explaining several aspects of community structure and increasing the likelihood of incorporating a wider range of responses to different stressors (*Karr, 1999*). However, this approach is dependent on metrics displaying different trajectories from each other, and the metrics used here all exhibited similar responses to each other (http://dx.doi.org/10.6084/m9.figshare.1053134). Thus, a much wider range of metrics than used in this study would benefit a multimetric approach, such as functional diversity or taxonomic distinctness (*Clarke & Warwick, 1998*).

Most of the variation explained in this study was by resource supply metrics, which likely reflects the fact that no anthropogenic stressors were present at these sites. The metrics used in this study are commonly applied in biomonitoring situations and have been demonstrated to respond to various anthropogenic stressors including land-use degradation, flow reduction and forest harvesting (*Lenat, 1988*; *James & Suren, 2009*; *Reid, Quinn & Wright-Stow, 2010*; *Shearer & Young, 2011*). Not surprisingly, given the lack of human influences that would increase nutrient or organic loads, the metrics with the weakest link with environmental drivers were those designed to respond to organic enrichments, namely MCI and QMCI.

## Overhead cover and resource supply metrics

Several environmental variables strongly influenced both macroinvertebrate metrics and multivariate structure. Of these, the most important were percent overhead canopy cover, percent bryophyte cover, stream-bed stability, debris jam cover, periphyton cover and biomass, and stream size. In a previous study on these streams, *Tonkin, Death & Collier (2013)* showed that disturbance and productivity were important drivers in open-but not closed-canopy streams, despite canopy cover not affecting periphyton biomass. Due to a high density of cover in many New Zealand forests, canopy cover often strongly influences stream communities due to lowered periphyton standing crops (*Winterbourn, 1990*; *Death & Zimmermann, 2005*). However, this forest presence does not necessarily shift the functional composition due to a comparatively low level of allochthonous material entering these streams (*Winterbourn, Rounick & Cowie, 1981*). Canopy clearance can, however, have far-reaching consequences and can follow the subsidy-stress pattern proposed by *Odum, Finn & Franz (1979)*; where the initial response is positive before a critical threshold is reached that leads to negative effects on biodiversity of stream biota (*Quinn, 2000*; *Allan, 2004*). This phenomenon may be related to a greater periphyton food supply for macroinvertebrates, in the absence of shading (*Death & Zimmermann, 2005*).

While we are dealing with pristine streams in the present study, lack of canopy is not usually an independent stressor, rather it is associated with complete shifts in land-use, which can come with much more intense pressures than simply clearing riparian vegetation (*Sponseller, Benfield & Valett, 2001*). Nonetheless, *Tonkin & Death (2012)* found similar relationships between productivity, disturbance and diversity when comparing streams from the region assessed in this study and those of a region dominated by intensive agriculture. As expected, there was a negative link between canopy cover percentage and

stream width in the present study, but this link was not as strong as one would expect. I sampled a range of stream sizes, from first- to sixth-order, but the only first order stream was spring-fed and the majority of second order streams had large spring inputs.

The weakness of this canopy-width link results from several of the smaller streams lacking riparian vegetation, or if present, the vegetation often consists of small shrubs and tussocks, as was evident with the strong correlation between catchment bare ground and tussock and overhead cover. The importance of canopy cover on these communities is, therefore, not a function of an underlying stressor such as land-use change, but a natural vegetation phenomenon. Riparian vegetation can structure stream communities in several ways, including provision of habitat, physical structure and resource supply to stream dwellers, as well as filtering land-use impacts (*Naiman & Henri, 1997*). The importance of canopy presence in this study, therefore, likely represents several other important habitat factors.

Along with overhead cover, bryophyte cover was an important driver of invertebrate communities in both the PLSR and BIO-ENV approaches, but it did not correlate with any invertebrate metric. Moss was clearly linked with stream stability and likely also reflects the flow regime of these sites. Nineteen of the sites had no moss and a further 13 had 5% cover or less, therefore, it is likely a non-linear effect. Many environmental factors drive non-linear threshold responses in ecological communities (*Dodds et al., 2010*), and while the PLSR was able to extract the most important drivers of community metrics, I cannot make any inference regarding non-linear effects because PLSR assumes linear relationships between the independent and dependent variables.

## Plantation forestry

The only variable to influence multivariate structure but not any of the metrics, other than abundance, was the percentage of plantation forestry in the riparian zone. Yet, despite being a local-scale measure, all plantation forestry sites were situated within one defined area consisting of mainly mature *Pinus radiata*. These sites could be simply comprised of a different suite of taxa, but not ultimately affect the invertebrate metrics such as taxonomic richness. However, previous studies have found similar macroinvertebrate communities between sites in mature *Pinus radiata* forestry and native forest in New Zealand streams (*Quinn et al., 1997*; *Quinn, Boothroyd & Smith, 2004*).

While a clear negative link with nMDS axis 1 was apparent for plantation forestry, this pattern was driven largely by one site with very low abundance, and thus forestry sites did not group separately from the remainder of sites in ordination space. Moreover, plantation forest percentage was not related strongly to any of the other environmental variables. In fact, removing this site from the analyses removed the influence of planted forest on both community structure and abundance. Thus it may be that this single site was inflating the influence of planted forest on these communities although, when excluded, the explanatory success of the models was reduced.

## CONCLUSIONS

Overall, the drivers of community metrics and multivariate structure were largely similar. Canopy cover and resource supply metrics were the most commonly identified environmental drivers in these pristine streams. For an area with little to no anthropogenic influence, substantial variation was explained in the macroinvertebrate community, with both uni- and multivariate approaches. Given the pristine condition of these streams, this finding highlights the importance of considering natural underlying environmental variation when assessing biodiversity against coarser gradients of environmental change. The strength of this link also highlights the considerable range of environmental conditions present in these pristine streams shaping the macroinvertebrate communities present. Despite most metrics performing in a similar manner, I suggest it is important to use a wide range of metrics and approaches to represent the entire macroinvertebrate community. Moreover, given the importance of canopy cover presence in driving these communities, as well as its potential to mask the effects of anthropogenic stressors, it is crucial to account for this variable, and further research its role, in bioassessment programs.

## ACKNOWLEDGEMENTS

I thank Roger Tonkin, Amber McEwan, Nicki Atkinson, Manas Chakraborty, Robert Charles, Logan Brown, and Alana Lawrence for help with fieldwork, and Mike Joy and Russell Death for assistance with site selection. Keith Wood at Ernslaw One Limited provided access to Karioi Forest sites. Department of Conservation allowed access to conservation areas. Kevin Collier, Esta Chappell, Svein Saltveit and two anonymous reviewers provided valuable comments on the manuscript.

### Funding

A Massey University Doctoral Scholarship supported JDT during sample collection. The funders had no role in study design, data collection and analysis, decision to publish, or preparation of the manuscript.

### Grant Disclosures

The following grant information was disclosed by the author:
Massey University Doctoral Scholarship.

### Competing Interests

The authors declare there are no competing interests.

### Author Contributions

- Jonathan D. Tonkin conceived and designed the experiments, performed the experiments, analyzed the data, wrote the paper, prepared figures and/or tables, reviewed drafts of the paper.

## Field Study Permissions

The following information was supplied relating to field study approvals (i.e., approving body and any reference numbers):

Department of Conservation, Turangi, New Zealand: TT-25903-FAU.

## Data Deposition

The following information was supplied regarding the deposition of related data:

Figshare:

Jonathan D. Tonkin (2014): Benthic macroinvertebrate data collected from 47 sites in the Tongariro National Park, New Zealand, Feb–April 2007.

http://dx.doi.org/10.6084/m9.figshare.979303

Jonathan D. Tonkin (2014): Physicochemical data collected from 47 sites in the Tongariro National Park, New Zealand, Feb–April 2007.

http://dx.doi.org/10.6084/m9.figshare.979302

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
