# Peer review of "Drivers of macroinvertebrate community structure in unmodified streams"

_PeerJ, doi:10.7717/peerj.465_

## Round 0.1 · original submission · Major Revisions

Please note that the editor originally assigned to this MS had to step down, so I was assigned to continue the process.

This is a well-written MS that describes substantial sampling on almost four dozen streams in New Zealand. The author also completed comprehensive analytical work on the data and the conclusions do not overreach the results.

The reviewers have suggested minor and major revisions. I am generally quite conservative in split decisions, so will also recommend major revisions. However, the revisions suggested are seemingly on the minor side of major.

Please carefully consider the various recommendations and complete, or rebut, them as you see fit.

I look forward to reading the author's revised MS and rebuttal. Thanks for submitting this work to PeerJ.

Reviewer 1 ·

Basic reporting

This article reports on uni and multivariate explanations for macroinverebrate diversity patterns in 47 near-natural streams in NZ. The reporing is basically ok, except more details should be present in the methods-- For example, stream width is not mentioned but comprises two full paragrpahs in the discussion. The author should probably note the table (table 1) of metrics etc in the methods. The author stresses the limited variability of the streams selected, but they actual comprise a high degree of variability-- I would highlight this instead of downplaying it (variabiltiy among pristine sites)- I think this would actually make the paper more interesting.

Experimental design

i like the basic idea presented in this study. My only real comment extends to the improtance of the 1 outlier in the dataset that seems to strongly influence all of the results. i think the author needs to rerun the analyses excluding this outlier and see if the results remain the same. I think there is a strong argument for not including this sample in the study. If northing changes, then a simple sentence stating this fact could be added and the comment disappears (for others reading the paper). lastly, as above- link the metric table in the methods to make sure readers are aware of all the information collected at each site.

Validity of the findings

The paper is concise and contains minimal speculation. maybe more precise statements concerning the regional vs local influences.. the subsidy-stress paragraph is interesting but the author needs to be careful to keep the 'pristine' side of he story clear and not get muddled with anthrogenic stress. basically, a good read in my opinion--

Additional comments

Other than the analysis issue stated above, i generally liked the idea being tested in this paper.

·

Basic reporting

When conducting research in polluted rivers, and when evaluating or grading the level of pollution in accordance to a.o. the Europen Water Framework Directive simple metrics are commonly used. Studies in pristine rivers/streams, evaluating the importance natural environmental variation assessing the impact of stressors on stream macroinvertebrate communities become important. However, I lack a description on how this can be done. As said in the conclusions; ”Moreover, given the importance of canopy cover presence in driving these communities, as well as its potential to mask the effects of anthropogenic stressors, it is crucial to account for this variable in assessment programs”. A topic for a future paper?

This is a valuable contribution, also well written. My only objection, but I think an important one, to this study is the lack of an evaluation fish and invertebrate predator might have on the benthic invertebrate fauna composition. Under Material and Methods, Study sites, there is no mentioning of fish. If fish are present at the sites the species need to be mentioned, also their main foraging strategy. It is important to have that information, because benthic feeding fish can in some cases control the density and composition of benthic invertebrates. Large sized invertebrate predators might also have a similar effect. This unlike drift feeders, like brown trout or salmon, were studies have revealed that they play no important structural role. So, please include fish in study site, a table?, and also make some comments on possible effects from fish if or when present at these sites.
If no fish, it need to be mentioned!

Line 196: Please indicate the meaning of “taxonomically rich” i.e. genius, species; also the meaning of “35 caddiesflies” ?, 22 true flies and so on. Is it no. of individuals or no. of species?

Also, “I” is in my opinion used to often in the context of this paper. A clear expression of this usage is seen when reading from line 89 to line 97. So please, rewrite some (most) of the sentences using “I”.

Experimental design

No comments

Validity of the findings

This is a valuable contribution, also well written, when evaluating or grading the level of pollution in rivers and streams. However, there is a lack in how the findings, i.e. the importance natural environmental variation as stressors on stream macroinvertebrate communities can be used when. assessing the effects of anthropogenic stressors. it is crucial to account for this variable in assessment programs”. A topic for a future paper?

Additional comments

See above

---

## Round 0.2 · accepted · Accept

Thank you for your comprehensive response to the reviewers' comments. You have addressed their concerns adequately, and this MS is now acceptable for publication. Thank you for submitting your research to PeerJ.